# Designing a Novel Peptide-Based Multi-Epitope Vaccine to Evoke a Robust Immune Response against Pathogenic Multidrug-Resistant *Providencia heimbachae*

**DOI:** 10.3390/vaccines10081300

**Published:** 2022-08-11

**Authors:** Muhammad Naveed, Mohsin Sheraz, Aatif Amin, Muhammad Waseem, Tariq Aziz, Ayaz Ali Khan, Mustajab Ghani, Muhammad Shahzad, Mashael W. Alruways, Anas S. Dablool, Ahmed M. Elazzazy, Abdulraheem Ali Almalki, Abdulhakeem S. Alamri, Majid Alhomrani

**Affiliations:** 1Department of Biotechnology, Faculty of Life Sciences, University of Central Punjab, Lahore 54590, Pakistan; 2Department of Microbiology, Faculty of Life Sciences, University of Central Punjab, Lahore 54590, Pakistan; 3Pak-Austria Fachhochschule, Institute of Applied Sciences and Technology, Mang, Haripur 22621, Pakistan; 4Department of Biotechnology, Faculty of Biological Sciences, University of Malakand, Chakdara 18800, Pakistan; 5Institute of Basic Medical Sciences, Khyber Medical University, Peshawar 25100, Pakistan; 6Department of Clinical Laboratory Sciences, College of Applied Medical Sciences, Shaqra University, Shaqra 15273, Saudi Arabia; 7Department of Public Health, Health Sciences College Al-Leith, Umm Al-Qura University, Makkah al-Mukarammah 24382, Saudi Arabia; 8Chemistry of Natural and Microbial Products Department, Pharmaceutical and Drug Industries Research Division, National Research Centre, Dokki, Giza 12622, Egypt; 9Department of Clinical Laboratory Sciences, Faculty of Applied Medical Sciences, Taif University, Taif 21944, Saudi Arabia

**Keywords:** gram-negative, antibiotic-resistance, whole proteome, MHC-I, immune-informatics, in-vitro testing

## Abstract

*Providencia heimbachae*, a Gram -ve, rod-shaped, and opportunistic bacteria isolated from the urine, feces, and skin of humans engage in a wide range of infectious diseases such as urinary tract infection (UTI), gastroenteritis, and bacteremia. This bacterium belongs to the Enterobacteriaceae family and can resist antibiotics known as multidrug-resistant (MDR), and as such can be life-threatening to humans. After retrieving the whole proteomic sequence of *P. heimbachae* ATCC 35613, a total of 6 non-homologous and pathogenic proteins were separated. These shortlisted proteins were further analyzed for epitope prediction and found to be highly non-toxic, non-allergenic, and antigenic. From these sequences, T-cell and B-cell (major histocompatibility complex class 1 and 2) epitopes were extracted that provided vaccine constructs, which were then analyzed for population coverage to find its reliability worldwide. The population coverage for MHC-1 and MHC-2 was 98.29% and 81.81%, respectively. Structural prediction was confirmed by validation through physiochemical molecular and immunological characteristics to design a stable and effective vaccine that could give positive results when injected into the body of the organism. Due to this approach, computational vaccines could be an effective alternative against pathogenic microbe since they cover a large population with positive results. In the end, the given findings may help the experimental vaccinologists to develop a very potent and effective peptide-based vaccine.

## 1. Introduction

The Enterobacteriaceae family includes *Providencia heimbachae*, an opportunistic pathogenic gram -ve oxidative rod-shaped bacteria. With no known human cases of Providencia until recently, it was initially discovered in the penguin excrement of healthy penguins in 1986 [1]. In 1983, *Providencia* was identified as two distinct species, *Providencia rustigianii* and *Providencia friedericiana*. Both species were characterized based on their molecular and biochemical characterization. After a complete analysis, it was proved that both species were different, and identified as new strains [2]. However, three known species, *P. haembachae, P. rettgeri,* and *P. stuarti,* exist that participate in different human diseases. Most of the species of *Providencia* are resistant to the existing antibiotic, which poses a threat to humans. The main cause of this resistance is due to their environmental condition and continuous exposure to an antibiotic [3]. According to recent studies, ten species of Providencia (*P. heimbachae, P. thailandensis, P. burhodogranariea, P. sneebia, P. vermicola, P. stuartii P. alcalifaciens, P. rustigianii, P. rettgeri,* and *P. huaxiensis)* are known to date [4]. Out of these ten species, three species, *P. heimbachae, P. rettgeri, and P. stuartii,* are clinically important because they take part in diverse types of infections [5]. These three strains are important due to their pathogenicity; *P. heimbachae* is the most important as its strains are already isolated from human cell lines. There are no identifiable strains of *P. rettgeri* and *P. stuartii* found in humans [6]. The three strains of *P. heimbachae* are ATCC 35613, STRAIN 9901, and NCTC12003 [7]. NTC12003 causes diarrhea in piglets and urinary tract infection (UTI) in humans, which can be life-threatening for humans [8].

The bacterial cell wall is especially important for its survival in its host. Most bactericidal antibiotics produce transpeptidase or beta-lactams, which break the cell wall and kill the bacterial cell. Environmental conditions and prolonged exposure to antibiotics cause them to become resistant to antibiotics, which is a major issue as most bacteria will become resistant to all antibiotics in the future, and then the treatment of bacterial diseases will become difficult. Bacteria develop their resistance by producing beta-lactamase enzymes, which inhibit the exposure of transpeptidase and beta-lactam-based antibiotics [8]. To avoid antibiotic resistance, chlorhexidine is an antiseptic agent which inhibits the division of bacteria of both kinds that can be gram-positive and negative. Its working mechanism is not completely understood, but it can be a remarkably effective way to avoid bacterial infection [9]. Antibiotic resistance is considered a public health emergency by the Food and Agriculture Organization (FAO), the World Organization for Animal Health (OIE), and the World Health Organization (WHO). The United Nations (UN) predicts that in 2019 700,000 people died due to antibiotic resistance, which may increase to 10 million by 2050. The purpose of this study is to design a potential vaccine candidate to avoid antibiotic resistance in bacteria and avoid infection that is caused due to their activity [10].

In recent studies, it is exceedingly difficult to approve new antibiotics, and bacteria have become resistant to many already existing antibiotics. In this regard, the recombinant vaccine is an alternative strategy with immense importance as it supplies complete information, as well as the chances of becoming resistant are exceptionally low. On the other hand, traditional vaccine design is a very costly, tedious process, and it requires more human trials to verify the vaccine, so it is comparatively easy to move towards genetically aided vaccines. Next-generation sequencing (NGS) and improved bioinformatics tools may be used to design such vaccines, which can be helpful in this regard [10].

The main approach of this study is to design a peptide-based multiple epitope vaccine to fight *Providencia heimbachae*. This approach can be extremely helpful in combating diseases caused by this bacterium. From the health care perspective, it can be said that vaccines can supply immunity against all sequenced strains of pathogenic bacteria because these vaccines are based on their core genome [11]. The vaccine architecture is built using the most antigenic, non-allergic, and non-toxic epitopes to improve its efficiency and immune reaction against that bacterium’s immunological responses. Millions of lives can be saved from life-threatening diseases. Moreover, such vaccines are economically important because they can be inserted into subjects with multiple carrier systems and are cheaper in price [3].

## 2. Materials and Methods

The research method used to construct a recombinant vaccine against *Providencia heimbachae* is given in Figure 1.

### 2.1. Retrieval of Complete Proteome of P. heimbachae

The complete proteome of *P. heimbachae* (strain ATCC 35613) has been retrieved from UniProt (UniProt ID: UP000078224) https://www.uniprot.org/proteomes/, accessed on 20 June 2022. All retrieved sequences are taken in FASTA format [12].

### 2.2. CD-Hit Analysis

Redundant and non-redundant proteins are present in the genome of bacteria. Redundant proteins can be represented many times in the whole proteome, which is not as important as a strong candidate for the vaccine. The CD-hit approach removes all unnecessary proteins, and non-redundant proteins are selected for further selection.CD-HIT web server is used for this purpose (http://weizhong-lab.ucsd.edu/cdhit-web-server/cgi-bin/index.cgi?cmd=cd-hit) accessed on 20 June 2022 [13].

### 2.3. Subcellular Localization

Proteins that are present on the surface are involved in pathogenicity, and these proteins are easily identified by host immune cells and create an immune response against them. Due to this pathogenicity, surface localized proteins can be considered strong candidates for vaccine design. This analysis was achieved by CELLO (http://cello.life.nctu.edu.tw/cello2go/) (accessed on 20 June 2022) a web server that is used for subcellular localization of proteins [14].

### 2.4. Virulent Protein Analysis

To find virulent proteins, the database which is used is the virulent factor database (VFDB). In this analysis, sub-cellular localized proteins are analyzed through the Basic Local Alignment Search tool for proteins (BLAST p) against the full proteome of that strain. Those strains that do not fulfill the criteria are discarded and the remaining are further processed [15].

### 2.5. BLAST p Analysis

BLAST p (https://blast.ncbi.nlm.nih.gov/Blast.cgi?PAGE=Proteins) (accessed on 20 June 2022)predicts the homology between selected pathogenic proteins and normal human flora. This analysis is performed to avoid any autoimmune response by the host cells [16] and CLUSTAL W is used to confirm that the selected epitopes are conserved within various Providencia species, which describes its effectiveness for other strains of the same specie (https://www.genome.jp/tools-bin/clustalw) (accessed on 20 June 2022).

### 2.6. Physiochemical Analysis

Physiochemical analysis of selected proteins gives some important properties (amino acid composition, theoretical PI molecular weight, instability index, atomic composition, estimated half-life, and GRAVY). This analysis was performed by Protparam (https://web.expasy.org/protparam/) (accessed on 20 June 2022), an online web server for physiochemical analysis [17]. Proteins were selected based on their molecular weight and instability index value with cut-off values of about 110 kDa and <40, respectively. The instability index, mentioned in Table 1, depicts the stability of the designed protein, usually in a test tube. If the index was below 40, the protein was said to be stable and fit for further analyses, if not, then it was assumed that the protein might not survive the in-vitro environment and be degraded.

### 2.7. Transmembrane Helices

To check the binding of selected protein in the cell membrane, transmembrane helices analysis was performed through an online web browser TMHMM v.2.0 (https://services.healthtech.dtu.dk/service.php?TMHMM-2.0) (accessed on 20 June 2022). Protein could only be a suitable candidate for vaccine deigning if its transmembrane helices value was still within the range of the cut-off value. Selected proteins were further analyzed [18].

### 2.8. Antigenicity Prediction

Antigenicity is defined by the ability of a foreign antigen to bind to the host immune cell and elicit an immunological response. Protein sequences with a high value of antigenicity were considered the strong vaccine candidate, but the cut-off was > 0.4. To identify the antigenicity VaxiJen 2.0 (http://www.ddg-pharmfac.net/vaxijen/VaxiJen/VaxiJen.html) (accessed on 20 June 2022) online tool was used [19].

### 2.9. Epitopes Prediction Phase

Antigens could be expected by using an online tool called the immune-epitope database (IEDB). Predicting B cell epitopes with the IEDB B cell epitope prediction tool (http://tools.iedb.org/bcell/) (accessed on 20 June 2022). MHC-2 and MHC-1 were used to predict T cell epitopes. The MHC-1 epitope could be predicted using an online tool. Epitope prediction for MHC-1 by the IEDB (http://tools.iedb.org/mhci/) (accessed on 20 June 2022), and the IEDB MHC-2 tool predicted the second class of MHC (http://tools.iedb.org/mhcii/) (accessed on 20 June 2022) [20].

### 2.10. Allergenicity and Toxicity Prediction

The vaccine may be allergenic or toxic for the host organism. To check allergenicity and toxicity AllerTOP (https://www.ddg-pharmfac.net/AllerTOP/) (accessed on 20 June 2022) and toxinpred (http://crdd.osdd.net/raghava/toxinpred/) (accessed on 20 June 2022) were used [21].

### 2.11. Multi-Epitope Vaccine Designing and Processing

All the antigenic epitopes screened for the multiple-epitope peptide-based vaccine were linked with the GPGPG linker. To design the vaccine construct, cholera toxin B adjuvant was added, enhancing the vaccine construct’s immunogenic effects [22].

### 2.12. Loop Modeling

Unnecessary loops were removed from the vaccine construct to obtain the vaccine structure. Loop modeling was performed through the online server GalaxyWEB server (https://galaxy.seoklab.org/) (accessed on 20 June 2022) [23].

### 2.13. Galaxy Refinement

A 3D structure after loop modeling was further analyzed to remove any side chain and unnatural overlaps between the protein molecules. The refined structure of the vaccine after remodeling was considered to be a good candidate for a vaccine, and it was carried through the galaxy refine server (https://galaxy.seoklab.org/cgi-bin/submit.cgi?type=REFINE) (accessed on 20 June 2022) [24].

### 2.14. Disulfide Engineering

The stability of vaccines could be enhanced by disulfide engineering, or their twisted structure was reduced in their conformational energy to obtain stability. Design 2.0 server (http://cptweb.cpt.wayne.edu/DbD2/index.php) (accessed on 20 June 2022) was used to analyze stability [25].

### 2.15. Codon Optimization

JCat (http://www.jcat.de/Literature.jsp) (accessed on 20 June 2022) tool multiple epitope sequences into DNA and then inserted DNA cloned inside *E. coli.* Expression analysis for this sequence was analyzed with the codon adaptation index (CAI) [26].

### 2.16. Molecular Docking

Molecular docking tested how well the vaccine and the immune cell receptors of the host organism stuck together. Molecular docking predicted how vaccines would bind to toll-like receptors (TLRs), major histocompatibility class 1 and 2 (MHC-1 and MHC-2, respectively) [27]. The basic principle of molecular docking was function scoring and sample confirmation. The fire dock server was used to complete this analysis. (https://bioinfo3d.cs.tau.ac.il/FireDock/php.php) (accessed on 20 June 2022) [28].

### 2.17. Molecular Dynamic Simulation (MDS)

Predicting the movement of atoms or molecules associated with vaccine construct stability using bioinformatics techniques, such as molecular dynamic simulation (MDS). For a brief time, vaccines and immune cells worked together. This interaction could be examined using the I-mod server. (https://imods.iqfr.csic.es/) (accessed on 20 June 2022) [29].

### 2.18. Immune Simulation

Immune simulation was used to predict the immune response generated against some antigens in the form of antibodies. It ensured all antibodies, cytokines, and interferons were generated by interacting antigens with host immune cells [30]. A tool used for this purpose was C-IMMSIM (https://www.iac.rm.cnr.it/~filippo/c-immsim/index.html) (accessed on 20 June 2022) [31].

## 3. Results

### 3.1. Retrieval of Proteomic Sequence, Subcellular Localization, and Transmembrane Alpha-Helices Identification

The complete proteomic sequence of *P. heimbachae* ATCC 35,613 was retrieved from the UniProt database and further analyzed for protein sequences that could be predicted for vaccine construct. Forty-two extracellular outer membrane pathogenic proteins were selected from the proteomic sequence. Non-homologous protein sequences from the human host were shortlisted for the vaccine, and homologous proteins were excluded using the BLASTp tool. For the next vaccine design approach, selected proteins were analyzed for allergenicity and antigenicity [32]. To check the allergenicity of selected proteins, AllerTop was used, and the antigenicity of those proteins was predicted by Vaxijen with the threshold level > 0.5; protein sequences with the high antigenicity, that were non-allergic, and non-toxic were separated, which could then be used to design multiple epitope vaccine construct (MEVC) [3].

### 3.2. Epitope Prediction and Population Coverage

B-cell epitopes could be predicted using the immunological epitope database. Multiple epitope vaccines were created by combining cytotoxic T-cell epitopes with helper T-cell epitopes. The immune response triggered by B-cells that is dependent on antibodies is known as a humoral immune response. A B-cell epitope prediction technique was used to predict B-cell epitope IEDB. Alternatively, T-cells supplied an immunological response at the cellular level; therefore, MHC-1 and MHC-2 epitopes were predicted from T-cell epitopes, and this method was used to predict T-cell epitopes. There were two methods used to predict MHC-1 and MHC-2, both of which used ANNs. A total of 12 B-cell epitopes, 11 MHC-1 epitopes, and 10 MHC-2 epitopes were found in this investigation. After predicting epitopes, the prevalence of MHC-1 and MHC-2 in populations around the world were examined to ensure the validity of vaccination designs. MHC-1 epitopes had a 98.29% population coverage, while MHC-2 epitopes had 81.811% population coverage, respectively.

### 3.3. Peptide-Based Vaccine Construction

Antigenicity, allergenicity, toxicity, physiochemical analysis, and solubility of the adjuvant MEVC were analyzed for each of the 12 B-cell, 11 MHC-1, and 10 MHC-2 epitopes linked together to construct a multi-epitope peptide-based vaccination against *P. heimbacha* as seen in Figure 2.

### 3.4. Structure Prediction and Validation

After getting the tertiary structure of the vaccine to construct, it was visualized in PyMOL as shown in Figure 2. The 2D structure of the vaccine was predicted by PSI-PRED software, as shown in Figure 3a. The 3D protein structure was predicted through Alpha fold Colab and visualized with the help of PyMol given in Figure 3b. The Galaxy web server further refined these three-dimensional structures; PROCHECK and ProSA were two bioinformatical tools that confirmed the validity of 3D vaccine constructed through the Z score value, which was 3.64; the structure could also be validated with the help of ERRAT by giving the quality factor, which is 99.6337. The Ramachandran plot represented the results of PROCHECK, which described the stability of protein with 95.64% efficacy, as shown in Figure 3c; the allowed and disallowed regions are expressed in Figure 3d).

### 3.5. Docking of Multiple Epitope Vaccine Construct

A BLAST and CLUSTAL W analysis of potential vaccine constructs showed that selected epitopes for the potential vaccine construct were effective for other strains of P. heimbachae. The docking analysis further evaluated the interaction between MEVC and human toll-like receptors 3 and 4 (TLR3 and TLR4). The tool used for this purpose was the Cluspro server. A representation of the docking analysis is given in Figure 4.

### 3.6. Insilco Cloning and Immune Simulation of Construct

The sole purpose of in silico cloning is to predict the accurate amplification of our potential vaccine candidate inside suitable micro-organisms. Jcat is a tool that improves the DNA sequence of vaccine construct to enhance CAI value and GC content; as a result, the optimized vaccine construct yields better results when they are subjected to expression analysis in *E. coli.* After that, the formed DNA fragments were incorporated into a vector for cloning. A C-ImmSim web server was used to check the immune response, which proved the immune response generated in the form of antibodies, as shown in Figure 5. Given results predicted that vaccines generate a strong immune response. The antibody production rate was 3 to 4; antibody production increased within 5 days and generated a strong immune response against pathogenic bacteria. Overall, the results showed that the potential vaccine candidate generated a strong positive immune response against bacteria.

### 3.7. Molecular Dynamic Simulation

MDS interprets the reliability of vaccines by interpreting the results in the form of atomic index graphs and B-factors. The I-Mod web tool used for MDS is given in Figure 5.

## 4. Discussion

*Providencia heimbachae*, is an opportunistic gram-negative bacterium that causes life-threatening diseases in humans and belongs to the family Enterobacteriaceae, which depicts resistance to drugs due to the production of beta-lactamase [33]. Initially, it is pathogenic for animals, but few species are reported in humans, which causes zoonotic infections [34]. This information raised the importance of computational vaccines against Providencia heimbachae; such vaccines are potentially effective, cost-effective, and have low chances for bacteria to develop antibiotic resistance [11].

The main concern of the study is to design a multi-epitope peptide-based vaccine against one of the most pathogenic bacterial spp. *P. heimbachae* [35]. A proteomic database was used to obtain the proteomic sequence and find homologous and non-homologous proteins that were analyzed for allergenicity and antigenicity to find the most compatible sequence for multiple epitope vaccines [36]. B and T cell epitopes are needed to design multiple epitope vaccines [37]. These epitopes are predicted from non-allergenic and highly antigenic protein sequences. These epitopes joined with each other through linkers and adjuvants to the multi-epitope vaccine construct [38]. To check potential effectiveness, molecular and dynamic simulation was completed since occasionally protein sequences of vaccine constructs are specifically for that pathogenic organism and non-homologous to proteins obtained from BLAST analysis [39].

The protein sequence can be a practical candidate for the vaccine if it is recognized by the body’s immune cells and surface exposed. To increase vaccine effectiveness, selected epitopes are analyzed for antigenicity and allergenicity by Vaxijen and Aller-TOP v2 [40]. These bioinformatics tools predict epitopes with the highest antigenicity, which increases the effectiveness. Additionally, B and T cells epitopes are combined since they generate humoral and cell-mediated immune responses, reducing the chances for pathogenic bacteria to become resistant from the potential vaccine candidate.

The result of this study was to design a multi-epitope peptide-based vaccine against the pathogenic bacteria *P. heimbachae* [41]. The total number of amino acids to design peptide-based vaccines was 578, which was highly antigenic, non-toxic, and non-allergenic. These constructs held 12 B-cell epitopes, 11 MHC-1, and 10 MHC-2 epitopes that started with cholera toxin B subunit adjuvant and were linked with each other through certain linkers (EAAAK, AAY, GPGPG) [42]. Linkers such as AAY and GPGPG were added between two distinct types of epitopes for the effective separation needed for the efficiency of the epitope, as shown in the work of Naveed et al. (2021) [43] and Naveed et al. (2022) [44]. At the N-terminal of the vaccine construct, an adjuvant was added to improve immunogenicity and vaccine delivery in the host. Another linker, EAAAK, was placed between the adjuvant and the first epitope to enhance the tertiary structure stability. The prime requirement of using these linkers was to keep the vaccine structure intact. Poly-histidine tags were added at the C-terminal of the vaccine to keep the structure stable [45]. The physicochemical analysis of the potential vaccine candidate evidenced it to be stable and suitable for clinical use. The vaccine construct of multiple epitopes was used to conduct population analysis, which predicted credibility of the vaccine worldwide. If the vaccine covered 60% of the population worldwide, it was considered dependable. Docking was completed with a human toll-like receptor (TLRs), which confirmed the interaction between the vaccine and the immune cells of the body; indirectly, it depicted the response generated when a vaccine was introduced into the body.

The effectiveness of vaccines was verified through the IEDB population coverage tool; the greater percentage of population covered, the greater the chances would be the effectiveness of the potential vaccine candidate. IEDB coverage values showed that the potential vaccine candidate gave more than 60% of the population in the world, so this could be a strong candidate for the potential vaccine candidate and confirmed all aspects from valid immunoinformatics and bioinformatics tools and produced long-lasting effects against pathogenic bacteria *P. heimbachae* and could be safe if it was carried towards lab trials [44]. Furthermore, it was also proven that the selected epitopes for the potential vaccine construct were conserved among the Providence species, improving its efficiency against all the strains of the species.

## 5. Conclusions

*P. heimbachae* is a life-threatening microbe for living organisms for two main reasons, as it causes infection in both animals and humans, as well as it is resistant to many antibiotics, which suggests it is extremely dangerous for living creatures. Therefore, designing a new drug is ineffective, due to the chances of it becoming resistant to that drug. Vaccine development as an alternative strategy could be remarkably effective due to the chances of resistance being less, and such computational vaccines are designed by considering all the related aspects and trustworthy tools. All the predicted epitopes are related to selected proteins, which participate in designing vaccine construct. All the epitopes are highly antigenic, non-allergic, and non-toxic with specific physicochemical characteristics, which indicate this vaccine a beneficial remedy against the selected bacterium. The potential vaccine candidate is safe if taken towards in-vitro and in-vivo trials.

## Figures and Tables

**Figure 1 vaccines-10-01300-f001:**
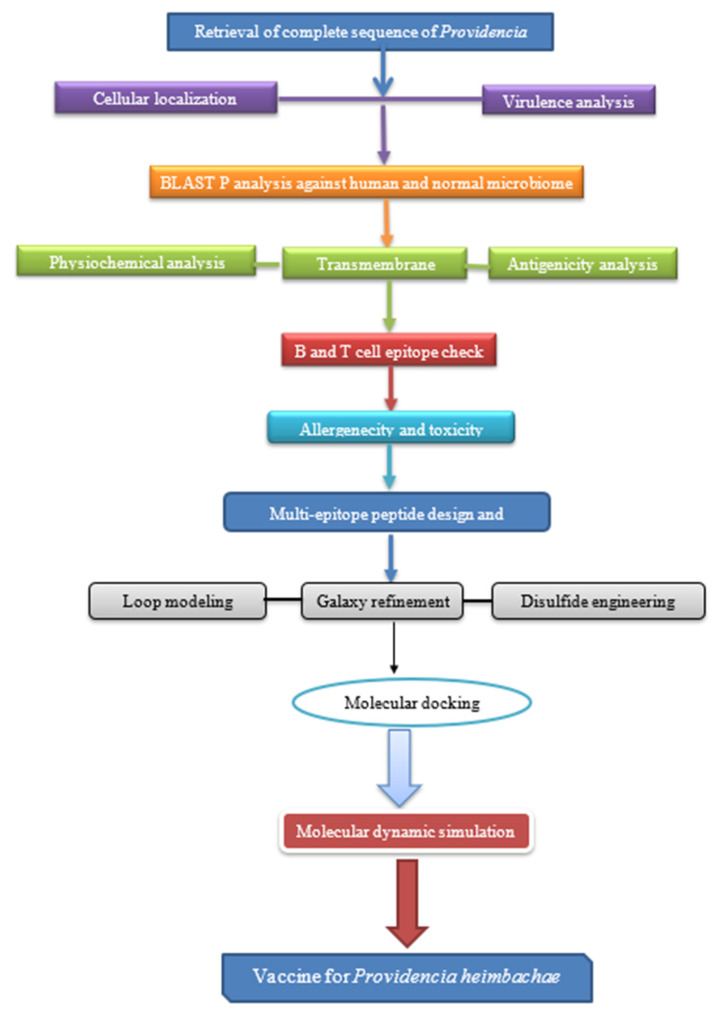
Stepwise representation of the complete method followed in this study. This method starts with proteomic sequence retrieval and finishes with peptide-based vaccine, followed by epitope prediction, secondary and tertiary structure prediction, refining molecular docking, and immune stimulation.

**Figure 2 vaccines-10-01300-f002:**
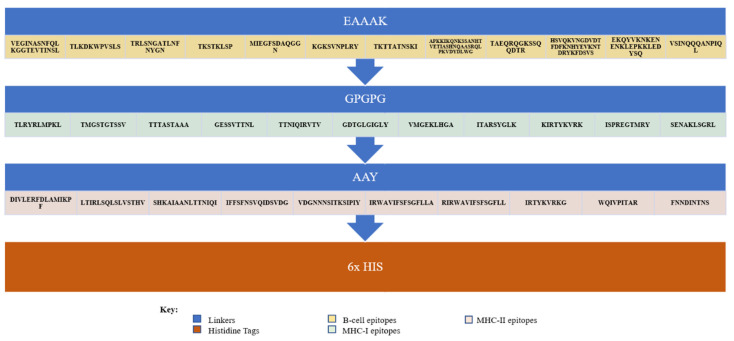
The vaccine construct, with linkers EAAAK in between the B-cell epitopes, GPGPG in between the MHC-I epitopes, AAY in between the MHC-II epitopes, and 6x His tag in the end.

**Figure 3 vaccines-10-01300-f003:**
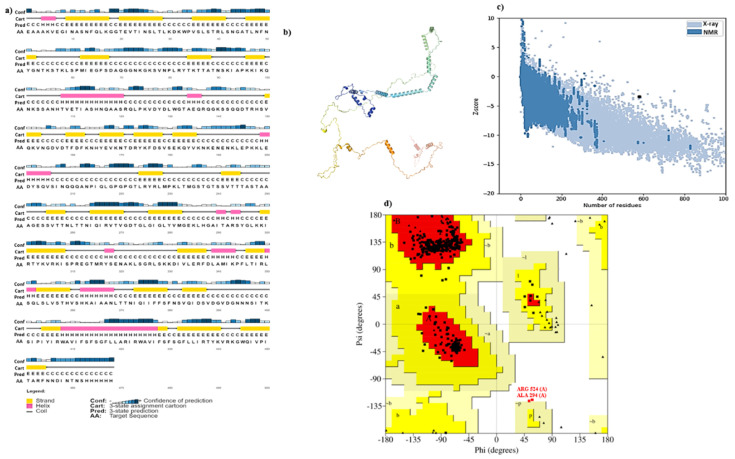
Structure prediction of the vaccine; (**a**) PSI-PRED predicted the 2D structure of the vaccine; (**b**) 3D structure predicted by Alpha fold Colab; (**c**) Z graph generated from ProSA web server; (**d**) Ramachandran’s plot describes the validity of vaccines through PROCHECK. The letters A and a represent right-handed alpha-helices (with A representing the region that has the most probability of being an alpha-helix and, a representing region with lesser probability and ~a with the least probability. The same goes for B, where B represents the region for B-sheets. L stands for left-handed alpha-helices with the probability decreasing from L to l. lastly ‘P’ represents the outlier region.

**Figure 4 vaccines-10-01300-f004:**
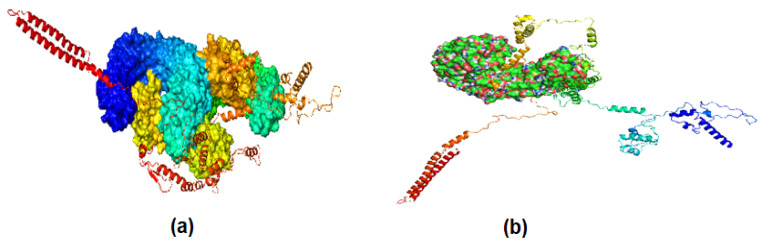
Molecular docking; (**a**) docking of vaccine constructs with TLR3; (**b**) docking vaccine construct of TLR4. The varying colors represents different chains of the interacting molecules.

**Figure 5 vaccines-10-01300-f005:**
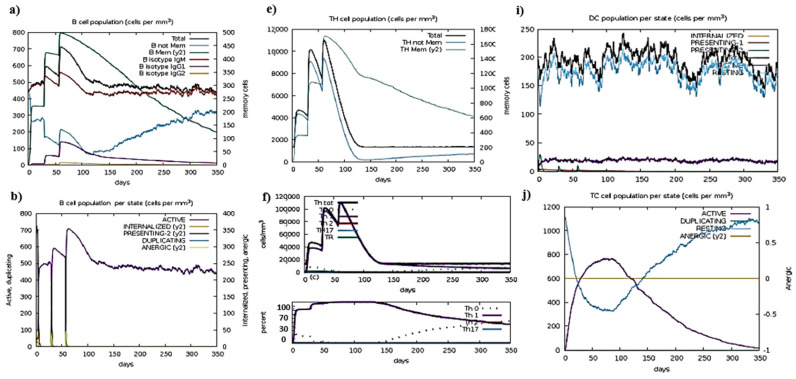
Representation of simulation; (**a**) After an antigen injection, the B cell population grows; (**b**) B cell population per state; (**c**) T helper cell population per state; (**d**) Immunoglobulin or antibody production after antigen injection; (**e**) T helper cell population after antigen injection; (**f**) T helper cell population in a cell in mass cell ratio with its percentage coverage; (**g**) T cell population after antigen interaction; (**h**) Natural killer cells population after antigenic vaccine interaction; (**i**) The population of dendritic cells as per state; (**j**) Population coverage of T cell as per state; (**k**) Population coverage of MA as per state; (**l**) The population of eosinophils per state after antigen injection.

**Table 1 vaccines-10-01300-t001:** Physiochemical analysis for multiple epitope vaccine construct (MEVC).

Property	Measurement	Indication
Total Number of Amino Acid	578	Appropriate
Molecular Weight	62067.34	Appropriate
Formula	C2761H4383N791O826S6	-
Theoretical pI	10.07	Basic
Total number of positively charged residues (Arg + Lys)	81	-
Total number of negatively charged residues (Asp + Glu)	35	-
Total Number of Atoms	8767	-
Instability index (II)	18.21	Stable
Aliphatic Index	68.58	Thermostable
Grand Average of Hydropathicity (GRAVY)	−0.558	Hydrophilic
Antigenicity VaxiJen	1.05	Antigenic
Allergenicity	Non-Allergen	Non-allergenic
Toxicity	Non-toxic	Non-toxic

## Data Availability

All major data generated and analyzed in this study are included in this manuscript.

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
