# Peer review of "Designing a Novel Peptide-Based Multi-Epitope Vaccine to Evoke a Robust Immune Response against Pathogenic Multidrug-Resistant Providencia heimbachae"

_vaccines, 2022, doi:10.3390/vaccines10081300_

Round 1

Reviewer 1 Report

Review comment on vaccines

An effective therapy against bacteria is always necessary in the modern world. This manuscript reported a new approach for constructing a multi-epitope vaccine against bacteria. For this purpose, an integrated approach that included epitope, allergenicity, toxicity, and antigenicity prediction was piped up to locate the multi-epitope vaccine. Many factors are known to influence the activity of a vaccine, including the allocation of epitope and flexibility of the branched chain. Despite inspiring work and excellent simulation results, experimental data to support the simulation is necessary. I strongly suggest that experimental data, the experimental data by chemicals, is needed to provide evidence of this work. Therefore, I do not recommend accepting this manuscript at this stage.  

Other concerns and suggestions

What is the purpose of in silico cloning and immune simulation?

In discussion, what are the "molecular and physicochemical properties"? I do not understand what the authors want to deliver.

I cannot entirely agree with this statement" all bactericidal antibiotics produce trans peptidase or beta-lactams.." Other categories of antibiotics did not make these two types of products. One representative example is an antimetabolite.

Minors

In table 2, what is the instability index? An appropriate explanation is necessary.

Refine English is suggested. There are several long sentences. Short ones are readable and understood. 

Author Response

An effective therapy against bacteria is always necessary in the modern world. This manuscript reported a new approach for constructing a multi-epitope vaccine against bacteria. For this purpose, an integrated approach that included epitope, allergenicity, toxicity, and antigenicity prediction was piped up to locate the multi-epitope vaccine. Many factors are known to influence the activity of a vaccine, including the allocation of epitope and flexibility of the branched chain. Despite inspiring work and excellent simulation results, experimental data to support the simulation is necessary. I strongly suggest that experimental data, the experimental data by chemicals, is needed to provide evidence of this work. Therefore, I do not recommend accepting this manuscript at this stage.  

Other concerns and suggestions

What is the purpose of in silico cloning and immune simulation?

AR: Thank you very much for your comment. The sole purpose of in silico cloning is to predict the accurate amplification of our potential vaccine candidate inside suitable micro-organisms, added to lines 279-280 in the revised manuscript and highlighted in red color. Immune simulation is used to check the predicted immune response generated by our potential vaccine construct as mentioned in lines 212-214. Please see revised manuscript.

In discussion, what are the "molecular and physicochemical properties"? I do not understand what the authors want to deliver.

AR: Thank you very much for your comments. Lines 341-343, has been changed the statement to ‘the physicochemical analysis of the potential vaccine candidate evidenced it to be stable and suitable for clinical use. Please see revised manuscript.

I cannot entirely agree with this statement" all bactericidal antibiotics produce trans peptidase or beta-lactams.." Other categories of antibiotics did not make these two types of products. One representative example is an antimetabolite.

AR: Thank you very much for your comments. The statement has been changed in the revised manuscript and highlighted in red color.

Minors

In table 2, what is the instability index? An appropriate explanation is necessary.

AR: Thank you very much for your comments. It has been explained in the revised manuscript please see line 149-153 highlighted in red color.

Refine English is suggested. There are several long sentences. Short ones are readable and understood. 

AR: Thank you very much for your comments. English language has been improved in the revised manuscript and long sentences have been removed. Please see revised manuscript.

Regards

Dr. Tariq Aziz (PhD, Postdoc)

Assistant Professor

Pak-Austria Fachhochschule Institute of Applied Sciences and Technology

Reviewer 2 Report

Authors show the design of a vaccine candidate against spp P. heimbachae using non-allergenic and highly antigenic protein sequences

Comments and recommendations

The manuscript should be reviewed, as there are some parts that are not understood e. g.: Lanes 81-82: The purpose of this study is to refrain from antibiotic resistance which can be acomplished by avoiding infections

The quality of all figures must be improved

It is recommended to use “potential vaccine candidate” and not emphasize that the designed construct is a vaccine, as this construct has not been evaluated in preclinical or clinical studies.

Discussion needs to be substantially improved

The authors do not explain what was the criterion for joining the sequences in the order shown in Figure 2. Were other constructs evaluated? If so, what was the criterion for selecting this construct?

What was the criteria for selecting the linkers EAAAK, AAY, GPGPG, SKK. Why did they use 4 linkers and not just one linker? What was the criterion for locating these linkers in the construct sequence? In addition to adjuvant properties, what other factors were taken into account?

Authors should explain whether the selected epitopes are conformational. If they are, how their location in the construct sequence may affect the induction of the immune response? If they are nonconformational epitopes, their location in the construct sequence would affect the induction of the immune response?

Authors should evaluate (e. g. using the Multiple Sequence Alignment – CLUSTALW) whether the selected sequences are located in conserved or variable regions of the proteins of the Providence species reported to date. What would be the implications for vaccine candidate efficacy if some of the selected sequences are from variable regions?

 Is the vaccine candidate effective only against resistant strains? Or if it is against all strains of P. heimbachae? This topic must be argued

Author Response

Authors show the design of a vaccine candidate against spp P. heimbachae using non-allergenic and highly antigenic protein sequences.

AR: Thank you very much for your comments.

Comments and recommendations

The manuscript should be reviewed, as there are some parts that are not understood e. g.: Lanes 81-82: The purpose of this study is to refrain from antibiotic resistance which can be acomplished by avoiding infections

AR: Thank you very much for your comments. It has been modified, corrected in the revised manuscript, and highlighted in red color. Please see revised manuscript.

The quality of all figures must be improved

AR: Thank you very much for your comments. The figures quality has been improved in the revised manuscript. Please see revised manuscript.

It is recommended to use “potential vaccine candidate” and not emphasize that the designed construct is a vaccine, as this construct has not been evaluated in preclinical or clinical studies.

AR: Thank you very much for your comments. It has been modified, corrected in the revised manuscript, and highlighted in red color. Please see line 81, 272, 280, 288-289, 328 356, 360, and 372- 373 in the revised manuscript.

Discussion needs to be substantially improved

AR: Thank you very much for your comments. Discussion has been improved in the revised manuscript and highlighted in red color. Please see revised manuscript.

The authors do not explain what the criterion was for joining the sequences in the order shown in Figure 2. Were other constructs evaluated? If so, what was the criterion for selecting this construct?

AR: Thank you very much for your comments. It has been corrected in the revised manuscript and highlighted in red color. Please see lines 333-336 in the revised manuscript.

What was the criteria for selecting the linkers EAAAK, AAY, GPGPG, and SKK. Why did they use 4 linkers and not just one linker? What was the criterion for locating these linkers in the construct sequence? In addition to adjuvant properties, what other factors were taken into account?

AR: Thank you very much for your comments. It has been corrected in the revised manuscript and highlighted in red color. Please see lines 336-343 in the revised manuscript.

Authors should explain whether the selected epitopes are conformational. If they are, how their location in the construct sequence may affect the induction of the immune response? If they are nonconformational epitopes, their location in the construct sequence would affect the induction of the immune response?

AR: Thank you very much for your comments. After the subcellular localization and ElliPro analysis all the selected epitopes lie on the outer membrane which depicts those epitopes are conformational and are directly involved in generating an immune response. Please see the relevant justification present in lines 124-129 highlighted in red color in the revised manuscript.

Authors should evaluate (e. g. using the Multiple Sequence Alignment – CLUSTALW) whether the selected sequences are located in conserved or variable regions of the proteins of the Providence species reported to date. What would be the implications for vaccine candidate efficacy if some of the selected sequences are from variable regions?

AR: Thank you very much for your comments. It has been corrected in the revised manuscript and highlighted in red color. Please see lines 139-142, 270-272 in the revised manuscript.

Is the vaccine candidate effective only against resistant strains? Or if it is against all strains of P. heimbachae? This topic must be argued.

AR: Thank you very much for your comments. It has been corrected in the revised manuscript and highlighted in red color. Please see lines 139-142, 359-361 in the revised manuscript.

Regards

Dr. Tariq Aziz (PhD, Postdoc)

Assistant Professor

Pak-Austria Fachhochschule Institute of Applied Sciences and Technology Haripur

Round 2

Reviewer 1 Report

I suggest accepting this manuscript in its current form. 

Reviewer 2 Report

The manuscript was satisfactorily corrected